# Accessibility and quality of care for adults with hypertension in rural Burkina Faso: results from a cross-sectional household survey

**Stephanie Lacey**[1]*, **Maria Lisa Odland**[1], **Ali Sié**[2], **Guy Harling**[3,4,5,6], **Till Bärnighausen**[7], **Pascal Geldsetzer**[8], **Lisa R. Hirschhorn**[9,10⚉], **Justine I. Davies**[1⚉]

**1** Institute of Applied Health Research, University of Birmingham, Birmingham, United Kingdom, **2** Centre de Recherche en Santé de Nouna, Nouna, Burkina Faso, **3** Institute for Global Health, University College London, London, United Kingdom, **4** MRC/Wits Rural Public Health and Health Transitions Research Unit (Agincourt), University of the Witwatersrand, Johannesburg-Braamfontein, South Africa, **5** Africa Health Research Institute, KwaZulu-Natal, South Africa, **6** University of KwaZulu-Natal, Durban, South Africa, **7** Heidelberg Institute of Global Health, University of Heidelberg, Heidelberg, Germany, **8** Department of Global Health and Population, Harvard T.H. Chan School of Public Health, Boston, Massachusetts, United States of America, **9** Department of Medical Social Science, Northwestern University Feinberg School of Medicine, Chicago, Illinois, United States of America, **10** Robert J. Havey Institute of Global Health, Northwestern University Feinberg School of Medicine, Chicago, Illinois, United States of America

⚉ These authors contributed equally to this work.
* stephanie.lacey@doctors.org.uk

**Data availability statement:** Due to privacy issues, data are not publicly available as consent was not given by participants for data to be shared openly. This is in part because entire age cohorts of some villages are included in the dataset, potentially allowing for deductive disclosure with sufficient local information. For this reason, anonymised data are available only following signature of a data use agreement

## Abstract

Providing quality healthcare is essential to reduce the future burden of cardiovascular disease. We assessed the quality of care for people with hypertension in Burkina Faso using the Institute of Medicine (IoM) Quality Domains of effectiveness, timeliness of access, patient-centredness and equitability of care. We performed an analysis of cross-sectional household survey data collected from a population-representative sample of 4000 adults ≥40 years in Nouna, Burkina Faso in 2018. For people with hypertension, effectiveness was assessed through care cascades describing the proportion who were screened, diagnosed, treated, and achieved hypertension control; timeliness was defined as access to care within the last three months. Patient-centredness was described using experiential quality process and outcome measures (dichotomised as higher or lower quality [score above or below and including the median, respectively]; a shared understanding and decision-making (SUDM) variable was described. Equity was assessed for effectiveness, timeliness, and patient-centredness in multivariable analyses including socio-demographic factors. In total, 1006 participants with hypertension were included. Hypertension prevalence was 34.8%; 62.3% had been screened, 42.9% diagnosed, 15.0% treated, and 6.8% were controlled; 26.8% had accessed care within the last three months. Overall, 61.8% of participants had a positive view of the health service. Clarity of communication and opinion of medical provider knowledge were the best-rated experiential quality process variables, with 40.1% and 39.7% of participants' responses indicating higher quality care respectively. The mean SUDM score was 68.5 (±10.8), range 25.0–100.0. Regarding equity, screening was higher in females, adults with any education, those who were

restricting onward transmission. Anyone wishing to replicate the analyses presented, or conduct further collaborative analyses using CHAS (which are welcomed and considered based on a letter of intent), should contact Sandra Barteit in the first instance. (Barteit@uni-heidelberg.de)

**Funding:** The CRSN Heidelberg Aging Study and author TB were supported by the Alexander Von Humboldt Foundation. Author TB received the Alexander von Humboldt professor award funded by the German Federal Ministry of Education and Research. Author GH was supported by a fellowship from the Wellcome Trust and Royal Society [grant number 210479/Z/18/Z]. The funders had no role in study design, data collection and analysis, decision to publish, or preparation of the manuscript.

**Competing interests:** The authors have declared that no competing interests exist.

married or cohabiting, and those in the higher wealth quintiles. There were no associations seen between SUDM and sociodemographic variables. Although the prevalence of hypertension was high in this population, the quality of care was not commensurate, with room for improvement in all four IoM Domains assessed.

## Introduction

Hypertension is the leading preventable cause of cardiovascular disease (CVD) and premature mortality globally [1,2]. A 2019 campaign that screened over 1.5 million adults from 92 countries worldwide estimated the prevalence of hypertension to be 34.0% [3]. Whilst the prevalence of hypertension is beginning to plateau or decrease in high-income countries (HICs), it is increasing in low- and middle-income countries (LMICs) [4]. As a result, the burden of disease disproportionately affects LMICs, with 80% of noncommunicable disease (NCD) related deaths occurring in these settings [5].

As a chronic disease, hypertension management requires patients to receive longitudinal quality care through the health system. However, simply being able to access care is not synonymous with receiving high-quality healthcare. Poor-quality care can lead to underutilisation or boycotting of formal health services as well as delays in access to care and is recognised as a key driver of preventable mortality [6–9]. Effective care for hypertension requires appropriate case-finding by screening for hypertension; ensuring people with hypertension have a formal diagnosis; offering effective treatment and counselling; and follow-up to monitor blood pressure control, adjust medications, and provide additional support as needed. However, multiple studies in LMICs show low awareness, treatment, and disease control [10]. Equity in access to effective hypertension care and achievement of equitable patient outcomes remains a key challenge globally [9,11,12].

A key determinant of whether patients remain in care and follow advice is patient-centredness, which describes care responsive to patient's preferences, needs and values whilst ensuring patients are involved in decisions about their care [13,14]. Larson et al. suggests that both process and outcome indicators should be used in determining patient-centredness of care and define the experience of care as a process indicator reflecting the interpersonal components of care received, with patient satisfaction, confidence in the health system, and met need as the outcome indicators of patient-centred care [15]. The responsiveness of care experience can be measured by asking patients to report their experiences of care (experiential quality) and assessing factors such as dignity, autonomy, confidentiality, and choice [14]. Results from surveys conducted in 49 LMICs from 2007 to 2017 found that on average, 34% of people from LMICs reported poor care experiences [9]. Shared decision-making is also an increasingly recognised important component of responsive care requiring collaboration between healthcare workers and patients in making decisions about disease management [14,16,17]. Higher-rated shared understanding and decision-making (SUDM) has been associated with greater trust and confidence in health services, as well as met needs [6].

Burkina Faso is a small, landlocked country located in Western Africa. Burkina Faso is a low-income country, and at the time this study was done, ranked 183rd out of 189 countries based on the Human Development Index [18]. Burkina Faso is undergoing a demographic transition, with increased life expectancy and falling birth rates resulting in a rapidly growing population [19,20]. Ischaemic heart disease is the fifth leading cause of death in Burkina Faso, only to be surpassed by malaria, respiratory infections, neonatal disorders and diarrheal diseases [20]. The World Health Organization (WHO) Stepwise approach to Surveillance (STEPS) survey conducted in 2013 estimated the prevalence of hypertension was 18.0% (95%

CI: 16.2–20.0) [21]. Despite this high prevalence, little is known about the quality of care for people with hypertension in Burkina Faso. Ensuring that the population receives the chronic disease management needed for this condition requires an understanding of the current status of care provided.

This analysis of cross-sectional data aimed to determine the quality of care for people with hypertension in a rural region of Burkina Faso based on selected healthcare quality domains of the United States Institute of Medicine (IoM) (now called the National Academy of Medicine) [13]. The main objectives were: i) to describe the population prevalence of hypertension and ii) to describe the quality of care for people with hypertension in terms of the IoM's domains of effectiveness, timeliness of access, patient-centredness, and equitability of access to effective, timely and patient-centred care.

## Methodology

### Study setting

This study was set in a predominantly rural area of Burkina Faso and based in the Nouna Health and Demographic Surveillance System (HDSS) area, which consists of the market town and 59 surrounding villages encompassing a total population of 107,000 people [6]. Farming constitutes the majority of economic activity in the region.

### Data collection

The HDSS is led by the Centre de Recherche en Santé de Nouna (CRSN). The CRSN partnered with the Heidelberg Institute of Global Health to conduct the CRSN Heidelberg Aging Study (CHAS). CHAS consisted of a cross-sectional household survey in 2018 of adults ≥40 years old living in the Nouna HDSS to explore health needs and health service utilisation of older adults living in Burkina Faso. Data collection methods are discussed in full elsewhere [22]. In brief, a stratified random population-representative sample of 4000 adults ≥40 years was obtained. Age-eligible participants were identified from the 2015 HDSS population census. In villages with more than 90 adults ≥40 years old, a random sample of households was generated, and one eligible adult was randomly selected from each selected household to participate. In villages with fewer than 90 eligible adults, all households with an adult ≥40 years were included. Data collection was conducted in participants' homes between 28th May and 16th July 2018 by trained data collectors. Written informed consent was obtained by study participants. In the case that the study participant was non-literate, the consent form was read to the participant and a thumbprint and literate witness was used to record consent. Interviews were conducted in French or another local language, and responses were recorded electronically on tablet devices using Open Data Kit Software.

### Definition of variables

**Demographic and disease characteristics.** Age was used as a continuous variable. Education level was characterised as none or any education. Marital status was categorised as married/cohabiting or single/separated/divorced/widowed. Household wealth was based on household assets and categorised in quintiles using the principal component method of Filmer and Pritchett [23]. In short, study participants were asked to complete survey questions related to household assets deemed indicative of wealth. Each asset was assigned a factor weight, and a wealth index score was calculated for each participant. Participant scores were ranked and divided into five equal groups (quintiles) (see S1 Table for further information on wealth quintile generation and household asset survey questions used) [24]. We included body mass index (BMI) in our analysis, given the hypothesised confounding effect of this

variable on the relationships between wealth and access to and experienced quality of care [25]. BMI in kg/m$^2$ was used to create categories of underweight (<18.5), normal weight (18.5 to 24.9), overweight (25.0-29.9), and obese (≥30.0). Hypertension was defined as a self-reported diagnosis of hypertension, and/or current receipt of treatment for hypertension, and/or investigator-measured systolic blood pressure ≥140 mmHg or diastolic blood pressure ≥90 mmHg.

**Quality of hypertension care.** Data were available to enable assessment of four of the IoM domains of quality healthcare: effectiveness, timeliness, patient centredness, and equitability. Data were not available to assess safety or efficiency of care.

Effectiveness of care was assessed through the construction of a care cascade. The prevalent population is the population defined as living with hypertension; screening was defined as the % of the population with prevalent hypertension who had ever had their blood pressure measured; diagnosis was defined as the % of the population with prevalent hypertension who had ever been told they have hypertension; treatment was defined as the % of the population with prevalent hypertension who were currently taking medication for hypertension; control was defined as the % of the population with prevalent hypertension whose blood pressure measured by study investigators was <140/90mmHg, as defined by the WHO's Package of Essential NCD Interventions (WHO PEN) [26]. Entry into the following step of the cascade was contingent on having achieved the previous one. The approach used in this study follows the same methodology used elsewhere [10,27,28]. Guidelines in Burkina Faso, state that people with hypertension should be in contact with the health services at least once every three months. Therefore, timeliness of access to care was described as the population with prevalent hypertension who attended a healthcare facility at least once in the last three months for any reason.

Patient-centredness of care was described using six experiential quality process variables, a composite of the SUDM variable, and three health system quality outcome variables. The six experiential quality process variables were: clarity of communication; ease of following instructions; opinion of medical providers knowledge and skills; trust in medical providers skills and abilities; involvement in decision making; and affordability of care. Following methodology described elsewhere [6], the SUDM variable was created as a composite using four variables which had the greatest factor loading in factorial analysis (clarity of communication, opinion of providers knowledge and skills, trust in medical providers skills and abilities, and involvement in decision making), and which the study investigators felt reflected the components necessary for SUDM. The three health system quality outcome variables were: met need from last visit; confidence in the health system; and overall view of the health system.

Equitability of access to effective, timely, and patient-centred care was defined as care that did not vary based upon a participant's sociodemographic characteristics which relate to equity (gender, age, education level, marital status, household wealth).

## Analysis

Results for continuous variables are described as mean (standard deviation [sd]) and range when normally distributed or median (and interquartile range [IQR]) when skewed. Categorical variables are presented as proportions. Complete case analysis was performed for all analyses.

For timeliness, we describe the proportion of the population with prevalent hypertension who attended a healthcare facility in the past three months for any reason. Given that experience of care varies with the context, we described experiential quality process variables and health system quality outcome variables based on relative ratings, with higher-quality

care defined as being a response above the median value for each variable and values below or equal to the median defined as lower-quality care. To identify the median value for each variable, histograms were drawn. The experiential quality process and health system quality outcome variables were then recoded to create the binary higher- and lower-quality care responses. For those people with hypertension who had ever accessed care, quality of care was described as the proportion of participants with hypertension that reported higher quality care at their last visit versus the proportion that reported lower quality care at last visit. The composite measure SUDM was described using mean (sd) and range. A subgroup exploratory analysis of patient-centredness of care (including experiential quality process variables, SUDM, and health system quality outcome variables) was performed including only the participants that accessed care in the last three months.

Multivariable analysis was done using binary logistic or linear regression, with dependent variables in each model as progress through the cascade, timeliness, or patient centredness variables. For each dependent variable two sequential models were performed. Model 1 adjusted for age, gender, education level, marital status, and wealth quintile; model 2 adjusted for the same variables as model 1 with the addition of BMI. Results for logistic regression models are presented as prevalence odds ratio (POR, equivalent to odds ratio). Results for linear regression models are presented as regression coefficients. Multicollinearity of independent variables was assessed by calculating the variance inflation factor (VIF). The VIF was <5 for all independent variables indicating a low likelihood of collinearity.

A p value of ≤0.05 was taken as representing statistical significance. All analyses were performed using STATA version 17.0.

### Ethical considerations

Ethical approval for the primary analysis was given from the Ethics Commission I (medical facility Heidelberg [S-120/2018]), the Burkina Faso Comité d'Ethique pour la Recherche en Santé (2018-4-045) and the Institutional Ethics Committee of the CRSN (2018-04). Written informed consent was provided by study participants and verbal approval was granted from village elders. The detection of undiagnosed hypertension in study participants led to a referral to an appropriate health service supported by provision of funds for transportation.

### Inclusivity in global research

Additional information regarding the ethical, cultural, and scientific considerations specific to inclusivity in global research is included in the Supporting Information (S1 Checklist).

## Results

In total, 3028 adults aged ≥40 years consented to participate in CHAS, of these, 2887 (95.3%) participants had complete data available (S1 Fig and S2 Fig). Of participants with hypertension, 91.8% (N=1006) had complete data available and were included in analyses. The prevalence of hypertension in people who had complete data was 34.8%. The sociodemographic characteristics of the prevalent hypertension population are show in Table 1.

**Effectiveness of care.** Among those with hypertension, 62.3% reported having previously had their blood pressure tested (screened for hypertension), with 42.9% having received a formal diagnosis, 15.0% being on treatment, and 6.8% having achieved controlled disease. The greatest loss in the care cascade occurred at the treatment and control stages, with 65.0% and 55.0% loss, respectively (Fig 1).

**Timeliness of access to care.** In total, 269 (26.8%) participants with hypertension sought care from a healthcare facility in the last three months (S2 Table).

**Table 1. Characteristics of the population with hypertension.**

| Parameter | Group | Overall population with hypertension N=1006 |
|---|---|---|
| | | N (%) |
| Gender | Male | 458 (45.5) |
| | Female | 548 (54.5) |
| Age, median (IQR) | – | 55 (47–64) |
| Education level | No formal education | 845 (84.0) |
| | Any education | 161 (16.0) |
| Marital status | Single/divorced/widowed | 292 (29.0) |
| | Married/cohabiting | 714 (71.0) |
| Wealth quintile | 1 | 199 (19.8) |
| | 2 | 164 (16.3) |
| | 3 | 182 (18.1) |
| | 4 | 213 (21.2) |
| | 5 | 248 (24.7) |
| BMI* | Underweight (<18.5 kg/m$^2$) | 142 (14.2) |
| | Normal range (18.5-24.9 kg/m$^2$) | 594 (59.3) |
| | Overweight (25-29.9 kg/m$^2$) | 178 (17.8) |
| | Obese (≥30-kg/m$^2$) | 88 (8.8) |

*Total population excludes four participants with missing BMI data. BMI, body mass index; IQR, interquartile range; N, number.

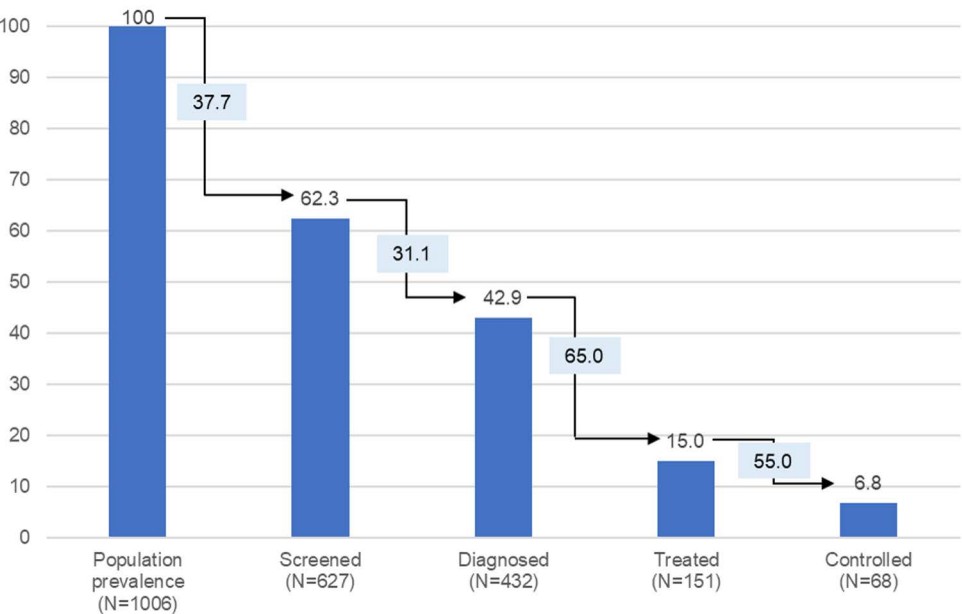

**Fig 1. Cascade of care for hypertension.** Bars represent the proportion (%) of the disease prevalent population progressing through each stage of the cascade. Boxes represent the proportion of participants lost at each stage of the cascade.

**Patient-centredness of care.** Of those with hypertension, 934 (92.8%) had complete visit (had responded to the question about whether they attended a health facility in the last three months or not) and experience of care responses. Experiential quality was categorised as higher quality in 11.1% of participants for ease of following instructions, in 40.1% of participants for clarity of communication, in 27.2% of participants for involvement in treatment, in 11.5% of participants for trust in skills and abilities of healthcare workers, and in 39.7% of participants for opinion of medical provider knowledge and skills. In total, 12.3% of participants had to borrow or sell something to pay for care. The mean SUDM score was 68.5 (±10.8), range 25.0 to 100.0 (Table 2).

Of the care quality outcomes, 44.5% of participants with hypertension reported that their need was met at last visit (excellent or good), 31.9% of participants were very confident if they became unwell tomorrow, they would receive effective treatment, and 61.8% of participants had a positive overall view of the health service (Table 2).

Of participants who had complete experience of care data available, only 250 (26.8%) participants had an appointment at a healthcare facility in the last three months and could be included in the subgroup analysis. When limiting the analysis to this subgroup, the proportion of participants reporting higher-quality care was greater than for those who had ever accessed care (S3 Table). Except for the domains of clarity of communication and need met at the last visit, whereby the proportion of participants reporting higher-quality care was numerically lower in participants with a recent visit (6.4% and 9.3%, respectively), compared with participants who had a healthcare facility appointment in the more distant past; statistical testing was not done.

**Equity.** Considering the effectiveness of care in multivariable analyses, the results from model 1 indicated that participants who were female, had any education, were married or cohabiting, or in wealth quintiles, 3 to 5 were more likely to be screened for hypertension (Table 3). Females were also more likely to receive a formal diagnosis of hypertension, and females or older adults were more likely to receive treatment for hypertension. There were no sociodemographic characteristics associated with disease control. There was an association between being in a higher wealth quintile and having increased odds of accessing care within the last three months, but no other sociodemographic characteristics were associated with timeliness (Table 4).

Results of the multivariable regression to examine the relationship between sociodemographic characteristics and experiential quality process measures in the population with prevalent hypertension are shown in S4 Table. Any education was positively associated with ease of following instructions, being male was associated with higher rating of clarity of communication, increasing age or residing in the 5th wealth quintile were positively associated with involvement in treatment decisions, and increasing wealth was associated with a reduced odds of having to borrow or sell things to afford care. There were no associations with any sociodemographic characteristics and SUDM (S5 Table). Considering health system quality outcomes measures in people with hypertension who had ever accessed care, people residing in the 3rd and 5th wealth quintiles had greater odds of reporting that the system met their needs. Still, there was no association with other sociodemographic characteristics on this outcome or any sociodemographic variable on confidence in or overall view of the healthcare system (Table 5).

Results for the exploratory analysis of associations between markers of equity and patient-centredness of care in the subgroup of participants with hypertension who sought care within the last three months are shown in S6 Table, S7 Table, and S8 Table. In brief, being in the 5th wealth quintile was associated with a lower rating of clarity of communication. There were no other significant associations between patient-centredness measures and participant characteristics in the subgroup analysis.

**Table 2. Patient-centredness of care for adults with hypertension.**

| Measure | Rating | N (%) | Binary rating | N (%) |
|---|---|---|---|---|
| Ease of following instructions* | Very easy** | 104 (11.1) | Higher quality | 104 (11.1) |
| | Easy | 692 (74.1) | Lower quality | 830 (88.9) |
| | Fair | 115 (12.3) | | |
| | Hard | 21 (2.2) | | |
| | Very hard | 2 (0.2) | | |
| Clarity of communication* | Excellent** | 57 (6.1) | Higher quality | 375 (40.1) |
| | Very good** | 318 (34.0) | | |
| | Good | 515 (55.1) | Lower quality | 559 (59.9) |
| | Fair | 41 (4.4) | | |
| | Poor | 3 (0.3) | | |
| Involvement in treatment decisions* | Excellent** | 40 (4.3) | Higher quality | 254 (27.2) |
| | Very good** | 214 (22.9) | | |
| | Good | 472 (50.5) | Lower quality | 680 (72.8) |
| | Fair | 129 (13.8) | | |
| | Poor | 79 (8.5) | | |
| Trust in skills and abilities of healthcare worker* | Very much** | 107 (11.5) | Higher quality | 107 (11.5) |
| | Quite a bit | 619 (66.3) | Lower quality | 827 (88.5) |
| | Some | 188 (20.1) | | |
| | Very little | 19 (2.0) | | |
| | Not at all | 1 (0.1) | | |
| Opinion of medical provider knowledge and skills* | Excellent** | 68 (7.3) | Higher quality | 371 (39.7) |
| | Very good** | 303 (32.4) | | |
| | Good | 514 (55.0) | Lower quality | 563 (60.3) |
| | Fair | 44 (4.7) | | |
| | Poor | 5 (0.5) | | |
| Shared understanding and decision making* (SUDM) (Range 25.0 to 100.0) | Mean (sd) | 68.5 (10.8) | | |
| Borrowed or sold anything to pay for healthcare* | Yes | 115 (12.3) | | |
| | No | 819 (87.7) | | |
| Overall, how well did received care meet health needs at last visit† | Excellent** | 80 (8.6) | Higher quality | 415 (44.5) |
| | Very good** | 335 (35.9) | | |
| | Good | 457 (49.0) | Lower quality | 517 (55.5) |
| | Fair | 49 (5.3) | | |
| | Poor | 11 (1.2) | | |
| Confidence that would receive effective treatment if very sick tomorrow† | Very confident** | 297 (31.9) | Higher quality | 297 (31.9) |
| | Somewhat confident | 567 (60.8) | Lower quality | 635 (68.1) |
| | Not very confident | 62 (6.7) | | |
| | Not at all confident | 6 (0.6) | | |
| Overall view of national health care system† | Only minor changes needed to healthcare system** | 576 (61.8) | Higher quality | 576 (61.8) |
| | Major changes needed to healthcare system | 324 (34.8) | Lower quality | 356 (38.2) |
| | Need to rebuild healthcare system | 32 (3.4) | | |

*Total population with hypertension included in the experiential quality analysis is 934 participants, this excludes those with missing visit or experiential quality data.

†Total population with hypertension included in the quality care outcomes analysis is 932 participants, this excludes participants with missing visit, experiential quality, and quality outcome data.

**Indicates responses above the median value and dichotomised as 'higher quality' for subsequent analyses. N, number; sd, standard deviation.

**Table 3. Multivariable association between sociodemographic characteristics and likelihood of progressing through the care cascade in the population with hypertension (model 1).**

| Parameter | Group | Screened vs not POR (95% CI) | P value | Diagnosed vs not POR (95% CI) | P value | Treated vs not POR (95% CI) | P value | Controlled vs not POR (95% CI) | P value |
|---|---|---|---|---|---|---|---|---|---|
| Gender | Male | Referent | – | Referent | – | Referent | – | Referent | – |
| | Female | **1.75 (1.28 to 2.39)** | **<0.001** | **1.60 (1.09 to 2.36)** | **0.017** | **1.71 (1.06 to 2.74)** | **0.027** | 1.40 (0.62 to 3.16) | 0.411 |
| Age* | | 1.01 (0.99 to 1.02) | 0.249 | 1.00 (0.99 to 1.02) | 0.750 | **1.03 (1.01 to 1.05)** | **0.002** | 0.99 (0.95 to 1.02) | 0.422 |
| Education level | No formal education | Referent | – | Referent | – | Referent | – | Referent | – |
| | Any education | **1.77 (1.14 to 2.73)** | **0.010** | 0.91 (0.58 to 1.43) | 0.674 | 1.12 (0.65 to 1.94) | 0.688 | 1.11 (0.45 to 2.72) | 0.823 |
| Marital status | Single/ divorced/ widowed | Referent | – | Referent | – | Referent | – | Referent | – |
| | Married/ cohabiting | **1.58 (1.09 to 2.27)** | **0.014** | 0.98 (0.61 to 1.58) | 0.943 | 0.89 (0.52 to 1.52) | 0.670 | 0.78 (0.33 to 1.86) | 0.575 |
| Wealth quintile | 1 | Referent | – | Referent | – | Referent | – | Referent | – |
| | 2 | 1.53 (0.99 to 2.34) | 0.053 | 1.35 (0.70 to 2.59) | 0.372 | 0.65 (0.29 to 1.47) | 0.301 | 2.31 (0.59 to 9.10) | 0.231 |
| | 3 | **1.92 (1.26 to 2.93)** | **0.002** | 1.16 (0.63 to 2.15) | 0.640 | 0.54 (0.25 to 1.19) | 0.125 | 0.54 (0.15 to 2.01) | 0.362 |
| | 4 | **2.94 (1.94 to 4.46)** | **<0.001** | 1.30 (0.73 to 2.33) | 0.376 | 0.91 (0.45 to 1.83) | 0.783 | 0.46 (0.15 to 1.42) | 0.175 |
| | 5 | **6.89 (4.33 to 10.95)** | **<0.001** | 1.54 (0.88 to 2.72) | 0.134 | 1.30 (0.67 to 2.55) | 0.436 | 1.00 (0.36 to 2.78) | 0.996 |

Model 1. Hypertension population includes those identified as hypertensive when examined by a study investigator and/or those who have previously been told they have hypertension and/or those who are currently taking treatment for hypertension (N=1006). The denominator population at each step of the cascade includes the participants that achieved the previous step. For example, the denominator population for the analysis of the association between sociodemographic characteristics and diagnosed vs not, are the participants that were screened for hypertension in the previous step of the cascade. Prevalent hypertension population (N=1006), screened (N=627), diagnosed (N=432), treated (N=151), controlled disease (N=68).

*Age in years, adults aged ≥40 years. CI, confidence interval; POR, prevalence odds ratio.

**Table 4. Multivariable regression shows the association between odds of timely access to care (healthcare appointment within the previous three months vs not) and participant characteristics representative of equity.**

**Model 1. Overall population with hypertension N=1004***

| Parameter | Group | POR 95% CI | P value |
|---|---|---|---|
| Gender | Male | Referent | – |
| | Female | 1.23 (0.89 to 1.70) | 0.202 |
| Age[†] | – | 1.01 (1.00 to 1.03) | 0.082 |
| Education level | No formal education | Referent | – |
| | Any education | 1.01 (0.67 to 1.52) | 0.958 |
| Marital status | Single/divorced/widowed | Referent | – |
| | Married/cohabiting | 0.93 (0.64 to 1.36) | 0.710 |
| Wealth quintile | 1 | Referent | – |
| | 2 | 1.13 (0.68 to 1.88) | 0.633 |
| | 3 | **1.75 (1.09 to 2.82)** | **0.021** |
| | 4 | **1.73 (1.09 to 2.73)** | **0.020** |
| | 5 | **1.74 (1.10 to 2.75)** | **0.018** |

*Overall population with hypertension excludes two participants with missing visit data.

[†]Age in years, adults aged ≥40 years. CI, confidence interval; N, number; POR, prevalence odds ratio.

**Table 5. Multivariable regression of the association between sociodemographic characteristics of participants with prevalent hypertension and health system quality outcomes.**

| Parameter | Group | Model 1 (N=932)* | | | | | |
|---|---|---|---|---|---|---|---|
| | | Reported met need | | Trust and confidence in health care system | | Overall view of the health care system | |
| | | POR (95% CI) | P value | POR (95% CI) | P value | POR (95% CI) | P value |
| Gender | Male | Referent | – | Referent | – | Referent | – |
| | Female | 0.97 (0.72 to 1.30) | 0.827 | 0.75 (0.55 to 1.02) | 0.065 | 0.75 (0.56 to 1.01) | 0.059 |
| Age† | – | 1.00 (0.99 to 1.01) | 0.802 | 0.99 (0.98 to 1.01) | 0.341 | 0.99 (0.98 to 1.01) | 0.362 |
| Education level | No formal education | Referent | – | Referent | – | Referent | – |
| | Any education | 1.01 (0.70 to 1.46) | 0.948 | 0.96 (0.65 to 1.41) | 0.833 | 0.81 (0.56 to 1.17) | 0.260 |
| Marital status | Single/ divorced/ widowed | Referent | – | Referent | – | Referent | – |
| | Married/ cohabiting | 1.00 (0.70 to 1.42) | 0.997 | 1.21 (0.82 to 1.79) | 0.331 | 0.85 (0.59 to 1.21) | 0.367 |
| Wealth quintile | 1 | Referent | – | Referent | – | Referent | – |
| | 2 | 1.44 (0.91 to 2.27) | 0.119 | 0.93 (0.56 to 1.54) | 0.767 | 0.98 (0.62 to 1.56) | 0.940 |
| | 3 | **1.56 (1.00 to 2.42)** | **0.050** | 1.03 (0.64 to 1.68) | 0.881 | 1.02 (0.65 to 1.60) | 0.928 |
| | 4 | 1.37 (0.89 to 2.09) | 0.149 | 1.44 (0.91 to 2.27) | 0.117 | 0.93 (0.61 to 1.43) | 0.745 |
| | 5 | **1.92 (1.26 to 2.93)** | **0.002** | 1.43 (0.91 to 2.25) | 0.117 | 0.86 (0.56 to 1.31) | 0.486 |

*Overall hypertension population (N=932) excludes participant with missing visit data, experiential quality data and quality care outcomes data.

†Age in years, adults aged ≥40 years. CI, confidence interval; N, number. POR, prevalence odds ratio.

Adding BMI to the models assessing equity of effectiveness, timeliness, and patient centredness for the main analysis in the prevalent hypertension population did not substantially change the results (S4 Table, S5 Table, S9 Table, S10 Table, and S11 Table). Regarding equity, when BMI was added to the model, being female was no longer associated with increased likelihood of receiving treatment for hypertension; being in the 3rd wealth quintile was no longer associated with reported met need; being in the 5th wealth quintile was associated with a higher rating of trust and confidence in the health system. Adding BMI into the models in the subgroup analysis of patients who had accessed care within the previous three months did not change the results, except for opinion of medical provider knowledge and skills, whereby increasing age was associated with higher rated quality of care in this domain.

## Discussion

In this study, the prevalence of hypertension among adults aged 40 and over was high (34.8%), yet within the range of earlier estimates for hypertension in Burkina Faso, ranging from 18.0–40.2% [21,29,30] and sub-Saharan Africa [27,31–34]. However, it is worth noting that earlier estimates in Burkina Faso included younger populations. Particularly the study by Soubeiga et al. 2017, which included adults aged 25–64 years and found the overall population prevalence to be 18.0% but as high as 37.3% in adults aged 55–64 years [21]. In a population with relatively low levels of overweight and obesity compared with many Western countries [35], the high prevalence of hypertension could be attributed to dietary behaviours related to salt intake. Salt intake is a key modifiable risk factor for hypertension and a recent study in Sierra Leonne found the prevalence of behaviours such as adding salt at the table or during cooking and eating salty snacks was high in adults ≥40 years [36].

We found that the effectiveness of care, as assessed by progression through the care cascade, was low with substantial dropout at the treatment and disease control stages. In adults

with hypertension, we found high levels of inequity in being screened at the first step in the cascade, with females, educated adults, married or cohabiting adults, and adults with higher household wealth all more likely to have reached this stage. Specifically, adults in the highest wealth quintile were almost seven times more likely to be screened than adults in the lowest wealth quintile. Whilst women remained more likely than men to have received a diagnosis and be on treatment, progression through other stages of the care cascade was similar in the other socio-demographic categories, with the exception that older adults were more likely to be on treatment for hypertension.

The fact that females are more likely to access and receive care for health conditions, including hypertension, has been described previously in high-, middle- and low-income countries [10,27,37]. Poor engagement and retention of men in healthcare could be due to the historic emphasis within global health on women's health, particularly sexual and reproductive health needs [38]. The gender equity gap in health service utilisation and need to understand male health-seeking behaviour is being increasingly recognised [38]. Unlike in some previous studies, we saw no significant association of marital status, education, or wealth with progression through the cascade beyond being screened [12,27]. This may be due to the lower number of participants in our study and the loss of power with transitioning through the cascade. However, these associations have been inconsistently shown in other studies, leaving open the possibility that the lack of association reflected reality [12,27].

Timely access to care in this study sample fell below national recommendations of access to care for people with hypertension, with only one-quarter of participants with hypertension accessing the healthcare service in the last three months. Lack of timely access to care for hypertension has been shown elsewhere in sub-Saharan Africa. In Tanzania, only 34% of adults with hypertension who were advised to seek healthcare attended a health facility in the subsequent 12 months [39]. In Sierra Leone, access to healthcare in the last three months for patients with CVD or CVD risk factors was low [27]. Infrequent engagement with healthcare services is problematic for hypertension management, which requires continuity of care including long-term medication adherence and follow up to prevent or delay complications including CVD events. Described barriers to accessing care are vast and include health system factors and patient-related factors, highlighting a need for context-specific exploration of barriers to care from the perspective of the patient and provider to develop effective interventions [40,41].

Affordability is an identified barrier to access to care and adherence to medication for hypertension [41]. In LMICs, the cost of healthcare can drain household income and may keep households in poverty as other necessities are foregone to pay for care. In our study wealth was associated with timeliness of access to care. While we did not examine out of pocket expenses or financial burden, we found the proportion of people with hypertension who ever accessed care who had to borrow or sell something in order to pay for a care episode was relatively small.

Continuity of care is essential to the effective management of NCDs. Whilst we did not assess continuity of care at the individual level, the low rates of progression at each stage of the care cascade and low proportion of participants adhering to local visit frequency guidelines suggests that continuity of care in this population was low. Novel approaches are being studied in LMICs such as the integration of NCD management into primary care [42]. Other work is exploring integration of NCD management into other programmes to improve comprehensiveness, coordination, and people centredness. For example, integrating hypertension screening and treatment into HIV clinics [43], although HIV is not a common condition in our population.

Assessing experiential quality measures is critical to understand the voice of the patient and inform where improvement may be needed to increase retention in and success with longitudinal care required for managing NCDs including hypertension [9]. Ratings of experiential quality measures will in part be influenced by cultural norms and expectations of care and thus creates an added challenge in interpreting responses. A low proportion of participants reported receipt of higher-quality care for the experiential quality process and most outcome measures, suggesting a need to improve the overall experience of care. The mean SUDM score in our study was 68.5 (±10.8), range 25.0–100.0. Another study which used the same dataset as in our study but included all participants who had accessed care (not limited to adults with hypertension), reported similar responses to care quality including SUDM [6]. The composite SUDM measure has not been used in any further studies to allow comparison of results. Improving patient experience of care is important as poor prior experiences of care can influence subsequent healthcare seeking behaviour [44]. In our study although the overall view of the health service was favourable, other experiential quality process measures suggest an underlying need to improve the quality of healthcare visits; meeting this need may improve the proportion of people accessing care in a timely manner in the future.

We found some inequity in receipt of patient-centred care mostly related to wealth, with those people with more assets having less need to borrow or sell anything to access care, reporting greater involvement in treatment decisions, and more likely to report the health system meeting their needs. It is possible that people who are wealthier may be treated differently to their less wealthy counterparts or are more empowered to engage in decision making, but this hypothesis requires further explanation.

The World Hypertension League has proposed an ambitious goal to be achieved by all African countries by 2030, which is that 80% of adults with hypertension should be diagnosed, 80% of those diagnosed (64% of hypertensive adults in total) should be on treatment and 80% of treated patients should have controlled disease [45]. They suggest a multi-dimensional approach to raise awareness, diagnosis and treatment of hypertension which involves action at a national policy level, healthcare provider and local community level. Our study results would suggest that the greatest gains in effectiveness of care would be to prioritise constrained resource on increasing the proportion of adults with diagnosed hypertension who are on treatment and have controlled disease. Whilst this approach focuses resource on a smaller cohort of adults with recognised hypertension and does not address the inequities of access to care further upstream, it at least has an impact on improving the CVD risk of some patients and hence future healthcare burden of CVD. Solutions to achieve this could include procuring low-cost single pill drug combinations, offering multi-month pill refills for those with stable disease, and decentralising healthcare and upskilling community health workers in hypertension management [45]. The latter may also enhance timeliness of access to care for hypertension and reduce the time and financial burden for obtaining care. In addition to this, there are some potentially low-cost population-level interventions aimed at reducing dietary salt intake that could help reduce the population prevalence of hypertension. These approaches include introducing regulations setting legal limits on added salt to processed food; introducing regulations on food labelling to include salt, sugar, and fat content per serving; and campaigns to raise public awareness of the importance of salt reduction [45]. In addition, improving identification of adults with hypertension could be enhanced through making blood pressure monitoring routine in all healthcare appointments and moving towards integrating care for multiple NCDs into single clinics to improve access [45].

Our study had a number of limitations. First, participants were labelled as having hypertension following a blood pressure measurement at a single encounter, whereas a clinical diagnosis requires raised blood pressure at two consecutive healthcare visits. Similarly, we relied

on self-report of prior diagnosis or being on treatment. Both issues may have led to an under or over reporting of diagnosed hypertension. However, these methods are often employed in epidemiological studies such as ours [10,27]. We did not have longitudinal data to fully understand aspects of care quality that might require a longer time horizon to capture, such as disease monitoring and patient follow-up. Understanding the experiential quality process and care outcome measures was limited to the patient's most recent interaction with the health service. Whilst cross-sectional study designs can identify associations between participant characteristics and quality of care measures, they cannot demonstrate causality. However, cross-sectional studies are frequently used to understand the current position of chronic disease management [10,28]. The small proportion of adults who received treatment or achieved disease control provided limited power to determine the relationship between experiential quality and progression in the care cascade. Similarly, the sample size of participants that sought care in the last three months was small and thus the results from the subgroup analysis should be interpreted with caution. Further, recall bias may have impacted the responses from those that didn't access care recently leading to a more positive recollection of the care experience. We did not use anchoring vignettes to adjust for how quality is rated differently in different countries based on prior experiences, norms and expectations of care [46]. Responses to the experience of care at last visit were not specific to appointments for hypertension due to the low number of participants who reported that the visit was for hypertension. We note that we do not present any formal assessment of the health system in this study and are therefore unable to present to the IoM domain of safety. When considering effectiveness, we also have not done a formal assessment of process measures such as provider competency or facility readiness to provide care (such as availability of medicines, affordability of care, workforce training, leadership, governance, record management, continuity of care); rather we have used clinical outcomes to assess if the health system is delivering effective outcomes. We also did not ask healthcare workers about their respect for patient values and preferences, as the outcome of importance was participants' experience of the health system. Some of the variables used in the multivariable analysis may be colinear, however, substantial co-linearity was excluded prior to conducting the multivariable analyses by ensuring the VIF was low.

## Conclusion

We found that quality of care for people with hypertension as determined by the IoM (currently National Academy of Medicine) domains of effectiveness, timeliness, patient-centredness, and equity in this population residing in rural Burkina Faso was low. Given that quality care is a central component of improving health outcomes, this study contributes to the global literature showing the breadth and depth of issues with quality care experienced in some low-income country settings.

## Supporting information

**S1 Fig.  Consort diagram for analysis model 1.** Model 1 did not include body mass index (BMI) in the multivariable analysis and thus includes participants with missing BMI data. (TIF)

**S2 Fig.  Consort diagram for analysis model 2.** Model 2 includes body mass index (BMI) in the multivariable analysis and thus removes participants with missing BMI data. (TIF)

**S1 Checklist.  Inclusivity in global research.** (DOCX)

**S1 Table. Household asset questions used to generate wealth quintiles using the Filmer and Pritchett principal components method.**
(DOCX)

**S2 Table. Description of the sociodemographic characteristics of participants with hypertension by timeliness of last attendance to a health facility.** *Overall population with hypertension excludes two participants with missing health facility visit data. †BMI excludes the four participants with missing BMI data. BMI, body mass index; IQR, interquartile range; N, number.
(DOCX)

**S3 Table. Patient-centredness of care for adults with hypertension that visited a health facility in the last three months (N=250).** *Above median values which equate to the 'higher quality' binary value. †One participant excluded for missing quality outcome data. N, number; sd, standard deviation.
(DOCX)

**S4 Table. Multivariable regression to determine the association between participant characteristics and experience of care in the prevalent hypertension population.** Model 1, prevalent hypertension population (N=934) excludes participants with missing visit or experiential quality data. Model 2 consists of the same population as model 1 except the four participants with missing body mass index data were removed. *Age in years, adults aged ≥40 years. BMI, body mass index; CI, confidence interval; N, number; POR, prevalence odds ratio.
(DOCX)

**S5 Table. Multivariable regression to determine the association between shared understanding and decision making (SUDM) and participant characteristics.** Model 1, prevalent hypertension population (N=934) excludes participants with missing visit or experiential quality data. Model 2 consists of the same population as model 1 except the four participants with missing body mass index data were removed. *Age in years, adults aged ≥40 years. BMI, body mass index; CI, confidence interval; N, number.
(DOCX)

**S6 Table. Multivariable regression to determine the association between participant characteristics and experience of care in the prevalent hypertension population who attended a healthcare facility in the last three months.** Model 2 excludes one participant with missing BMI data. *Age in years, adults aged ≥40 years. †27 participants dropped from analysis. BMI, body mass index; CI, confidence interval; N, number; POR, prevalence odds ratio.
(DOCX)

**S7 Table. Multivariable regression to determine the association between participant characteristics and shared understanding and decision making (SUDM) in participants with hypertension who attended a healthcare facility in the last three months.** Model 2 excludes one participant with missing BMI data. *Age in years, adults aged ≥40 years. BMI, body mass index; CI, confidence interval; N, number.
(DOCX)

**S8 Table. Multivariable regression to show the association between health system quality outcomes and participant sociodemographic characteristics in participants with hypertension that accessed care in the last three months.** *Model 1 and model 2 exclude one participant with missing health system quality outcome data. †Age in years, adults aged ≥40 years. **27 participants dropped from reported met need analysis. BMI, body mass index; CI, confidence interval; N, number. POR, prevalence odds ratio.
(DOCX)

**S9 Table. Multivariable association between sociodemographic characteristics and likelihood of progressing through the care cascade (model 2).** Model 2 Hypertension population includes those identified as hypertensive when examined by a study investigator and/or those who have previously been told they have hypertension and/or those who are currently taking treatment for hypertension, excludes participants with missing BMI data (N=1004). The denominator population at each step of the cascade includes the participants that achieved the previous step. For example, the denominator population for the analysis of the association between sociodemographic characteristics and diagnosed vs not, are the participants that were screened for hypertension in the previous step of the cascade. Prevalent hypertension population (N=1002), screened (N=623), diagnosed (N=428), treated (N=149), controlled disease (N=68). *Age in years, adults aged ≥40 years. BMI, body mass index; CI, confidence interval; N, number; POR, prevalence odds ratio.
(DOCX)

**S10 Table. Multivariable regression shows the association between odds of timely access to care (healthcare appointment within the previous three months vs not) and participant characteristics representative of equity (model 2).** *Overall population with hypertension excludes four participants with missing BMI data. †Age in years, adults aged ≥40 years. BMI, body mass index; CI, confidence interval; N, number; POR, prevalence odds ratio.
(DOCX)

**S11 Table. Multivariable regression to show the association between health system quality outcomes and participant sociodemographic characteristics in participants with hypertension regardless of timeliness of access to care (model 2).** *Two participants excluded due to missing health system quality outcome data. †Age in years, adults aged ≥40 years. BMI, body mass index; CI, confidence interval; N, number; POR, prevalence odds ratio.
(DOCX)

## Acknowledgements

For the purpose of open access, the author has applied a CC BY public copyright licence to any Author Accepted Manuscript version arising from this submission.

## Author contributions

**Conceptualization:** Ali Sié, Guy Harling, Till Bärnighausen.

**Formal analysis:** Stephanie Lacey.

**Methodology:** Pascal Geldsetzer, Justine I Davies.

**Supervision:** Maria Lisa Odland, Lisa R Hirschhorn, Justine I Davies.

**Writing – original draft:** Stephanie Lacey.

**Writing – review & editing:** Stephanie Lacey, Maria Lisa Odland, Ali Sié, Guy Harling, Till Bärnighausen, Pascal Geldsetzer, Lisa R Hirschhorn, Justine I Davies.

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
