## [Decision Letter · Decision Letter 0]

1 Jul 2024

PGPH-D-24-00560

Accessibility of care and experiential quality of care for adults with hypertension in rural Burkina Faso: results from a cross-sectional household survey.

Dear Dr. Lacey,

Thank you for submitting your manuscript to PLOS Global Public Health. After careful consideration, we feel that it has merit but does not fully meet PLOS Global Public Health’s publication criteria as it currently stands. Therefore, we invite you to submit a revised version of the manuscript that addresses the points raised during the review process.

We look forward to receiving your revised manuscript.

Kind regards,

Dr Buna Bhandari

Academic Editor

Journal Requirements:

Additional Editor Comments (if provided):

Reviewers' comments:

Reviewer's Responses to Questions

**Comments to the Author**

1. Does this manuscript meet PLOS Global Public Health’s publication criteria? Is the manuscript technically sound, and do the data support the conclusions? The manuscript must describe methodologically and ethically rigorous research with conclusions that are appropriately drawn based on the data presented.

Reviewer #1: Yes

Reviewer #2: Yes

2. Has the statistical analysis been performed appropriately and rigorously?

Reviewer #1: Yes

Reviewer #2: Yes

3. Have the authors made all data underlying the findings in their manuscript fully available (please refer to the Data Availability Statement at the start of the manuscript PDF file)?

Reviewer #1: No

Reviewer #2: Yes

4. Is the manuscript presented in an intelligible fashion and written in standard English?

Reviewer #1: Yes

Reviewer #2: Yes

5. Review Comments to the Author

Reviewer #1: This is an excellent manuscript covering an important topic for improving the services provided to patients with hypertension in Burkina Faso. The manuscript is well written and statistical analyses are robustly done. There are few very minor areas that I would like to suggest to authors to look at as follows:

65 Introduction

 Line 105: Write WHO in its long form and the abbreviation in brackets

 Line 115: Edit “IOMs” by adding an apostrophe between M and s so that it will be as follows: IOM's

Methodology

142 Definition of variables

143 Demographic and disease characteristics

 Line 147 (categorised in quintiles using the principal component method of Filmer and Pritchett. We): I suggest to the authors to add a reference for Filmer and Pritchett in order to assist a reader(s) to get more details as may be needed

155 Quality of hypertension care

 Line 166 (investigators was <140/90mmHg, as defined by the World Health Organisations Package of): I suggest to the authors to replace “World Health Organisations” with the WHO's

340 Discussion

 Line 421 (We found some inequity in experiential quality. Mostly related to wealth, with those people): I suggest to the authors to EDIT the sentence between the two words (…..quality. Mostly…..) by removing full stop and the word "Mostly" to be with a small "m"

472 Conclusion

 Line 473 (We found that quality of care for hypertension as determined by the Institute of Medicine): I suggest to the authors to replace “Institute of Medicine” with the following: IOM (currently National Academy of Medicine).

Reviewer #2: While the manuscript broadly meets the basic criteria for publication in PLOS Global Public Health, it needs to be improved before final publication.However, I have several concerns regarding the study's design and analysis methodologies that need to be addressed before the manuscript can be considered for publication.

6. PLOS authors have the option to publish the peer review history of their article (what does this mean?). If published, this will include your full peer review and any attached files.

**Do you want your identity to be public for this peer review?** For information about this choice, including consent withdrawal, please see our Privacy Policy.

Reviewer #1: **Yes: **Eliudi Saria Eliakimu

Reviewer #2: No

---

## [Decision Letter · Decision Letter 1]

30 Oct 2024

PGPH-D-24-00560R1

Accessibility and quality of care for adults with hypertension in rural Burkina Faso: results from a cross-sectional household survey.

Dear Dr. Lacey,

Thank you for submitting your manuscript to PLOS Global Public Health. After careful consideration, we feel that it has merit but does not fully meet PLOS Global Public Health’s publication criteria as it currently stands. Therefore, we invite you to submit a revised version of the manuscript that addresses the points raised during the review process.

We look forward to receiving your revised manuscript.

Kind regards,

Dr Buna Bhandari

Academic Editor

Journal Requirements:

Additional Editor Comments (if provided):

Reviewers' comments: Please see the attached document

Reviewer's Responses to Questions

**Comments to the Author**

1. If the authors have adequately addressed your comments raised in a previous round of review and you feel that this manuscript is now acceptable for publication, you may indicate that here to bypass the “Comments to the Author” section, enter your conflict of interest statement in the “Confidential to Editor” section, and submit your "Accept" recommendation.

Reviewer #1: All comments have been addressed

Reviewer #2: All comments have been addressed

2. Does this manuscript meet PLOS Global Public Health’s publication criteria? Is the manuscript technically sound, and do the data support the conclusions? The manuscript must describe methodologically and ethically rigorous research with conclusions that are appropriately drawn based on the data presented.

Reviewer #1: Yes

Reviewer #2: Yes

3. Has the statistical analysis been performed appropriately and rigorously?

Reviewer #1: Yes

Reviewer #2: Yes

4. Have the authors made all data underlying the findings in their manuscript fully available (please refer to the Data Availability Statement at the start of the manuscript PDF file)?

Reviewer #1: No

Reviewer #2: (No Response)

5. Is the manuscript presented in an intelligible fashion and written in standard English?

Reviewer #1: Yes

Reviewer #2: Yes

6. Review Comments to the Author

Reviewer #1: (No Response)

Reviewer #2: Please see the attached summary review.

7. PLOS authors have the option to publish the peer review history of their article (what does this mean?). If published, this will include your full peer review and any attached files.

**Do you want your identity to be public for this peer review?** For information about this choice, including consent withdrawal, please see our Privacy Policy.

Reviewer #1: **Yes: **Eliudi Saria Eliakimu

Reviewer #2: **Yes: **Deus M Bazira

---

## [Decision Letter · Decision Letter 2]

16 Jan 2025

Accessibility and quality of care for adults with hypertension in rural Burkina Faso: results from a cross-sectional household survey.

PGPH-D-24-00560R2

Dear Dr Lacey,

We are pleased to inform you that your manuscript 'Accessibility and quality of care for adults with hypertension in rural Burkina Faso: results from a cross-sectional household survey.' has been provisionally accepted for publication in PLOS Global Public Health.

Best regards,

Julia Robinson

Executive Editor

Reviewer Comments (if any, and for reference):

Reviewer's Responses to Questions

**Comments to the Author**

1. If the authors have adequately addressed your comments raised in a previous round of review and you feel that this manuscript is now acceptable for publication, you may indicate that here to bypass the “Comments to the Author” section, enter your conflict of interest statement in the “Confidential to Editor” section, and submit your "Accept" recommendation.

Reviewer #1: All comments have been addressed

Reviewer #2: All comments have been addressed

2. Does this manuscript meet PLOS Global Public Health’s publication criteria? Is the manuscript technically sound, and do the data support the conclusions? The manuscript must describe methodologically and ethically rigorous research with conclusions that are appropriately drawn based on the data presented.

Reviewer #1: Yes

Reviewer #2: Yes

3. Has the statistical analysis been performed appropriately and rigorously?

Reviewer #1: Yes

Reviewer #2: Yes

4. Have the authors made all data underlying the findings in their manuscript fully available (please refer to the Data Availability Statement at the start of the manuscript PDF file)?

Reviewer #1: No

Reviewer #2: Yes

5. Is the manuscript presented in an intelligible fashion and written in standard English?

Reviewer #1: Yes

Reviewer #2: Yes

6. Review Comments to the Author

Reviewer #1: (No Response)

Reviewer #2: Considering the study design and data collected, the authors have done the most they could.

7. PLOS authors have the option to publish the peer review history of their article (what does this mean?). If published, this will include your full peer review and any attached files.

**Do you want your identity to be public for this peer review?** For information about this choice, including consent withdrawal, please see our Privacy Policy.

Reviewer #1: **Yes: **Eliudi Saria Eliakimu

Reviewer #2: No
